

# A new model of meteoric calcium in the mesosphere and lower thermosphere

John M. C. Plane[1], Wuhu Feng[1,2], Juan Carlos Gómez Martín[1,3], Michael Gerding[4], Shikha Raizada[5]

[1] School of Chemistry, University of Leeds, Leeds LS2 9JT, U.K.
[2] National Centre for Atmospheric Science, School of Earth and Environment, University of Leeds, Leeds LS2 9JT, UK.
[3] Instituto de Astrofísica de Andalucía (IAA-CSIC), 18008 Granada, Spain.
[4] Leibniz Institute of Atmospheric Physics, Rostock University, Schlossstraße 6, 18225 Kühlungsborn, Germany.
[5] Space and Atmospheric Science Department, Arecibo Observatory/SRI International, Arecibo 00612, Puerto Rico.

*Correspondence to*: John M.C. Plane (j.mc.plane@leeds.ac.uk)

**Abstract.** Meteoric ablation produces layers of metal atoms in the mesosphere and lower thermosphere (MLT). It has been known for more than 30 years that the Ca atom layer is depleted by over 2 orders of magnitude compared with Na, despite
these elements having essentially the same elemental abundance in chondritic meteorites. In contrast, the $Ca^+$ ion abundance is depleted by less than a factor of 10. To explain these observations, a large data-base of neutral and ion-molecule reaction kinetics of Ca species, measured over the past decade, was incorporated into the Whole Atmosphere Community Climate Model (WACCM). A new meteoric input function for Ca and Na, derived using a chemical ablation model that has been tested experimentally with a Meteoric Ablation Simulator, shows that Ca experiences significant differential ablation, by almost 1
order of magnitude, with respect to Na. WACCM-Ca simulates the seasonal Ca layer satisfactorily when compared with lidar observations, but tends to overestimate $Ca^+$ measurements made by rocket mass spectrometry and lidar. A key finding is that CaOH and $CaCO_3$ are very stable reservoir species because they are involved in essentially closed reaction cycles with $O_2$ and O. This has been demonstrated experimentally for CaOH, and in this study for $CaCO_3$ using electronic structure and statistical rate theory. Most of the neutral Ca is therefore locked in these reservoirs, enabling rapid loss through polymerization into
meteoric smoke particles and explaining the extreme depletion of Ca.

## 1 Introduction

Layers of metal atoms and ions occur in the Earth's mesosphere and lower thermosphere (MLT, between 70 and 120 km) as a result of meteoric ablation (Plane et al., 2015). A layer of $Ca^+$ ions was first observed in the twilight glow more than 60 years ago (Vallance-Jones, 1956). Seven years later, Istomin (1963) used rocket-borne mass spectrometry to show that these ions
occurred in a broad layer peaking around 100 km. A number of sounding rocket measurements of metallic ions in the MLT have been made subsequently (Kopp, 1997a; Grebowsky and Aikin, 2002). The neutral Ca layer was first observed in 1985 by



the resonance lidar technique, and shown to peak around 90 km (Granier et al., 1985). Since then, a relatively small number of lidar studies of Ca and $Ca^+$ (which uniquely among the meteoric metal ions can be observed by ground-based lidar) have been performed (Alpers et al., 1996; Gerding et al., 2000; Granier et al., 1989; Qian and Gardner, 1995; Raizada et al., 2011; Raizada et al., 2012).

When compared to the Na and Fe layers, which have been much more widely studied, three features stand out (Plane, 2011). First, the atomic Ca abundance is depleted by a factor of 100 - 200, depending on season, with respect to Na, even though they have essentially the same abundance in carbonaceous Ivuna (CI) chondrites (Lodders, 2003), which are thought to be closest in composition to interplanetary dust particles (IDPs) (Jessberger et al., 2001). Second, the ratio of $Ca^+$ to Ca is around 10:1, whereas the analogous ratios for Na and Fe are around 0.2:1 (Plane et al., 2015). Third, whereas the Na and Fe layers at mid-
to high latitudes have marked seasonal variations, with a wintertime maximum more than 3 times larger than the summertime minimum, there appears to be much less seasonal variation in the Ca layer (Gerding et al., 2000).

There have been two modeling studies, published nearly two decades ago, which attempted to explain these striking differences between Ca and the other meteoric metals (McNeil et al., 1998; Gerding et al., 2000). In order to account for the low relative abundance of Ca, these studies assumed that Ca ablates much less efficiently than Na from IDPs, based on a thermodynamic
model of high-temperature melts (Fegley and Cameron, 1987). The reaction kinetics of most of the relevant neutral and ion-molecule reactions that Ca-containing species are likely to undergo in the MLT had not yet been studied, and so had to be estimated from the analogous reactions of other metals – whereas the second and third features of the Ca layer highlighted in the previous paragraph would indicate unexpected differences in chemical behavior. Although the modeling study of Gerding et al. (2000) achieved reasonable agreement with the annual average Ca layer measurements reported in the same paper (at
Kühlungsborn, Germany, 54° N), the seasonal variation of the layer and the $Ca^+$/Ca ratio were not modeled satisfactorily.

Since then, we have carried out a number of kinetic studies of the neutral chemistry of Ca and its oxides ($CaO$, $CaO_2$, $CaO_3$), hydroxides ($CaOH$ and $Ca(OH)_2$), oxy-hydroxides ($OCaOH$ and $O_2CaOH$) and carbonate ($CaCO_3$) with atmospherically relevant species such as $O_3$, O, H, $O_2$, $H_2O$ and $CO_2$ (Campbell and Plane, 2001; Plane and Rollason, 2001; Broadley and Plane, 2010; Gómez-Martín and Plane, 2017). We have also studied the relevant ion-molecule chemistry of $Ca^+$, $CaO^+$, $CaO_2^+$
and various cluster ions (Broadley et al., 2007; Broadley et al., 2008), including the dissociative recombination of $CaO^+$ with electrons (Bones et al., 2016a). This chemistry is illustrated in Fig. 1, which uses blue-shaded boxes to indicate ionized Ca-containing species, and green boxes for the neutrals. The ion-molecule chemistry is broadly similar to that of Fe (Plane et al., 2015). However, the neutral chemistry contains two significant features. First is the formation of the relatively stable carbonate, $CaCO_3$. This has been shown experimentally to react very slowly with atomic O, consistent with a large energy barrier that
has been calculated theoretically (Broadley and Plane, 2010). In Section 2 we discuss further why formation of $CaCO_3$ is most probably a sink for calcium.  Second, CaOH is involved in a holding cycle with $O_2CaOH$ and $OCaOH$, which substantially slows down its rate of its conversion back to Ca (Gómez-Martín and Plane, 2017). These two features indicate that atomic Ca may be depleted because of conversion into very stable reservoir species.



In another development, we have recently constructed a Meteoric Ablation Simulator (MASI) to study experimentally the ablation of different metals from meteoritic fragments, under heating conditions that simulate atmospheric entry (Bones et al., 2016b). Work with the simulator has confirmed that Ca does indeed ablate much less efficiently than Na from meteoritic particles, and allowed the Chemical Ablation Model (CABMOD) to be refined and validated (Gómez-Martín et al., 2017).

CABMOD could then be applied to the size and velocity distribution of IDPs entering the Earth's atmosphere – from Jupiter Family comets, asteroids, and Long Period comets – to determine the meteoric input functions of Ca and Na (Carrillo-Sánchez et al., 2016). This showed that the integrated injection rate of Na was 8.7 times larger than that of Ca i.e. nearly 1 order of magnitude larger than the CI ratio of the metals.

The objective of the present study is therefore to combine the substantial improvement in the data-base of Ca reaction kinetics with the experimentally-based Meteoric Input Function (MIF) for Ca in a global chemistry-climate model, and to determine whether the severe depletion of atomic Ca and the unusually large $Ca^+$/Ca ratio can now be explained.

## 2 Methods

### 2.1 Ca Chemistry

The rate coefficients for the reactions illustrated in Fig. 1 are listed in Table 1. Most of these have now been measured, as indicated in the footnotes to the table. As mentioned in the Introduction, we have recently carried out an experimental study of CaOH associating with $O_2$, followed by sequential reactions with atomic O to yield CaOH via $O_2CaOH$ and $OCaOH$ (Gómez-Martín and Plane, 2017). Here we use electronic structure calculations at the B3LYP/6-311+g(2d,p) level of theory (Frisch et al., 2009) to show that $CaCO_3$ should be involved in an analogous holding cycle, illustrated in Fig. 2a:

$$CaCO_3 + O_2 \ (+M) \ \rightarrow \ O_2CaCO_3 \qquad \Delta H^o = -89 \text{ kJ mol}^{-1} \qquad (R40)$$

$$O_2CaCO_3 + O \ \rightarrow \ OCaCO_3 + O_2 \qquad \Delta H^o = -110 \text{ kJ mol}^{-1} \qquad (R41a)$$

$$OCaCO_3 + O \ \rightarrow \ CaCO_3 + O_2 \qquad \Delta H^o = -311 \text{ kJ mol}^{-1} \qquad (R42)$$

Reaction R40 should be very fast in the MLT, because $O_2$ is a major atmospheric constituent and the reaction has a large calculated rate coefficient of $k_{40}(200 \text{ K}) = 4.0 \times 10^{-26} \ (T/200)^{-3.85} \text{ cm}^6 \text{ molecule}^{-2} \text{ s}^{-1}$ (see the Supplemental Information, SI). R41 involves reaction with atomic O, which is another major species above the "atomic O shelf" around 82 km (Plane et al., 2015). As shown in Fig. 2b, this reaction can produce $OCaCO_3$ (R41a) on a triplet electronic surface via transition state TS1 or on a singlet surface via TS2 (molecular parameters are listed in the SI). The highly exothermic reaction R42 involving atomic O then reduces $OCaCO_3$ back to $CaCO_3$ to complete the cycle.

However, R41 can also produce $O_2CaO_2$ on the singlet surface via TS3:

$$O_2CaCO_3 + O \ \rightarrow \ O_2CaO_2 + CO_2 \qquad \Delta H^o = -65 \text{ kJ mol}^{-1} \qquad (R41b)$$





which would be followed by $O_2CaO_2$ reacting with a second O:

$$O_2CaO_2 + O \rightarrow CaO_3 + O_2 \qquad \Delta H^o = \text{-311 kJ mol}^{-1} \qquad \text{(R43)}$$

As shown in Fig. 1, $CaO_3$ is chemically labile, with pathways to other oxides and hydroxides, and so if R41b is competitive with R41a then $CaCO_3$ (and $O_2CaCO_3$) would be a less stable calcium reservoir. We therefore performed a Rice-Ramsperger-
Kassel-Markus (RRKM) calculation on the reaction $O_2CaCO_3 + O$ using the Master Equation Solver for Multi-Energy well Reactions (MESMER) program (Glowacki et al., 2012; Robertson et al., 2012). The molecular parameters are listed in Table S3. The MESMER calculation shows that at $T = 200$ K and $p = 10^{-5}$ bar (typical conditions of the $80 - 85$ km region), then 99.6% of the reaction product is $OCaCO_3$ (produced in essentially equal amounts on the triplet and singlet surfaces), and 0.4% is $O_2CaO_2$. Thus the $CaCO_3 \rightarrow O_2CaCO_3 \rightarrow OCaCO_3 \rightarrow CaCO_3$ cycle should sequester most of the calcium below the peak
of the atomic Ca layer. The rate coefficient $k_2$ should also be around $2 \times 10^{-10}$ cm$^3$ molecule$^{-1}$ s$^{-1}$, essentially equal to the capture rate because the barriers on the potential energy surfaces are submerged below the entrance channel (Fig. 2b). Because the e-folding lifetime of $CaCO_3$ against recombination with $O_2$ is only ~10 ms at a height of 82 km, and the e-folding time of R42 should be around 0.25 s, ~96% of the $CaCO_3$ will be clustered to $O_2$.

Reaction R22 in Table 1 is a polymerization reaction, which describes the permanent loss of the neutral reservoir species to
form meteoric smoke particles. We have used this type of reaction previously for modeling the Na and Fe layers (Feng et al., 2013; Marsh et al., 2013a). Here, the rate coefficient $k_{22}$ is set to $9 \times 10^{-8}$ cm$^3$ s$^{-1}$. This is around 100 times larger than a typical dipole-dipole capture rate for these metallic molecules, which often have very large dipole moments (e.g. the calculated $\mu_D$ for $CaCO_3$ is 13.5 Debye (Broadley and Plane, 2010)). The reason for increasing the rate coefficient is that the Ca reservoir species can also polymerize with other (i.e., non-Ca containing) meteoric molecules (e.g., $NaHCO_3$, $FeOH$, and $Mg(OH)_2$), and the
dimerization rate coefficient needs to be increased to account for this since Ca ablates in a large excess of these other metals. A similar procedure was used to model the K layer, where the dimerization rate coefficient of $KHCO_3$ was increased by a factor of 270 (Plane et al., 2014).

The photo-ionization rate of Ca (R38 in Table 1) in the MLT was first estimated by Swider (1969). Using the more recently measured cross section of 2.0 Mb near the threshold at 202.9 nm (Ahmad et al., 1994), and 52 Mb at the 188.6 nm peak in the
photo-ionization spectrum (McIlrath and Sandeman, 1972), we obtain essentially the same photo-ionization rate as Swider of $5 \times 10^{-5}$ s$^{-1}$. Note that Ca has an unusually fast photo-ionization rate, which is 2.5 times faster than Na and 100 times faster than Fe (Plane et al., 2015).

## 2.2 Whole Atmosphere Model of Ca

The 3D global model of meteoric calcium was constructed by adding the Ca chemistry described in Section 2.1 into the Whole
Atmosphere Community Climate Model (WACCM). WACCM uses the framework from the fully coupled global climate model Community Earth System Model (CESM) (Hurrell et al., 2013), and is a comprehensive numerical model extending



vertically from the Earth's surface to the lower thermosphere (~140 km) (e.g. Marsh et al. (2013b)). For the present study we used a specific dynamics (SD) version of WACCM discussed in Feng et al. (2017), which is nudged with NASA's Modern-Era Retrospective Analysis for Research and Applications (MERRA) (Lamarque et al., 2012) below 60 km, and has the fully interactive chemistry described in Marsh et al. (2013b). The horizontal resolution is 1.9° latitude × 2.5° longitude, with 88

vertical model layers giving a height resolution of ~3.5 km in the MLT. In order to compare the Ca with the Na layer, a WACCM-Na simulation was also performed, where the Na chemistry was updated from Marsh et al. (2013a) with the results of two recent kinetic studies from our laboratory (Gómez-Martín et al., 2016; Gómez-Martín et al., 2017).

The injection profiles of Ca and Na used in WACCM-Ca and WACCM-Na are illustrated in Fig. 3. These Meteoric Input Functions (MIFs) were determined for the input of cosmic dust particles from three sources: Jupiter Family Comets, the

asteroid belt, and Long Period Comets (i.e. Halley Type and Oort Cloud Comets) (Carrillo-Sánchez et al., 2016). The elemental ablation rates of individual particles, selected using a Monte Carlo procedure from the dust size and velocity distributions predicted by an astronomical model (Nesvorný et al., 2011), were processed through CABMOD (Vondrak et al., 2008). These rates were then summed over the size/velocity distributions to produce the MIF (Carrillo-Sánchez et al., 2016).

It has become apparent in the past few years that global models such as WACCM underestimate the transport of minor species

through the MLT. This appears to be because the chemical and dynamical transport caused by dissipating atmospheric gravity waves can exceed transport driven along mixing ratio gradients by the turbulent eddy diffusion produced by breaking waves (Gardner et al., 2016). Short wavelength waves are not resolved within the coarse horizontal grid scales of models such as WACCM (~300 km resolution), so that much of the wave spectrum causing dynamical/chemical transport is not captured (in contrast, eddy diffusion caused by the breaking of sub-grid scale waves is parameterized). In the absence of these additional

vertical transport components, the MIF needs to be reduced in order for the model to produce the observed metal density. In the case of the Na MIF from Carrillo-Sánchez et al. (2016), a reduction by a factor of 5 produces good agreement with observations (see Section 3). Because a primary aim of this study is a comparison between Ca and Na, the Ca MIF was reduced by the same factor. These reduced MIFs are plotted in Fig. 3. The seasonal variation of these MIFs with latitude (Fig. S2) was derived by scaling to the variation of the Na and Fe MIFs determined previously using an astronomical dust model (Marsh et

al., 2013a; Feng et al., 2013).

Three model simulations were performed. The first run was WACCM-Ca from 1996 to early 2000 (termed the standard run), covering the period when the Ca and Ca$^+$ lidar measurements were made at Kühlungsborn, Germany (54.1°N, 11.7°E) (Gerding et al., 2000). The second model run was WACCM-Ca and WACCM-Na from 2004-2014, to obtain a 10-year climatology which also covers the period when lidar measurements were made at Arecibo, Puerto Rico (18°N, 293°E) (Raizada et al.,

2011). The third run was a sensitivity experiment, where the rate coefficients $k_{24}$ and $k_{25}$ were changed from their experimental values to their ion-molecule capture rate coefficients, calculated using Langevin Theory (Smith, 1980) (see footnote to Table 1). The reason for doing this is discussed in Section 3. The model outputs for the lidar stations at Kühlungsborn and Arecibo,



and the rocket launching sites at Kiruna, Sweden (68°N, 22°E), Redlake, Canada (51°N, 267°E) and Wallops, Virginia, U.S. (38°N, 285°E) were sampled every 30 minutes as in Feng et al. (2013).

## 2.3 Observational data

The lidar soundings at Kühlungsborn (54°N) were carried out with the double resonance lidar system described by Alpers et
al. (1996) and Gerding et al. (2000). With this lidar system two dye lasers are operated in parallel, allowing for simultaneous, common-volume soundings of Ca and $Ca^+$ (or Ca/Na or Ca/Fe, depending on configuration). For this study we extended the data set described by Gerding et al. (2000). Overall, 131 nights of Ca observations were recorded between December 1996 and January 2000. Many of these show prominent sporadic layers (e.g. Gerding et al. (2001)). These nights are excluded here, because they are strongly related to sporadic electron layers and other parameters that are not currently simulated in WACCM.
75 nights of observations remain (i.e., 42 nights of observations in 1997, 24 nights in 1998, 7 nights in 1999 and 2 nights in January 2000), each covering between 0.5 h and 10.3 h. In order to compare the model most directly with the observations when determining the seasonal variation of Ca, the model output was sampled at the times of the individual Ca measurements. The second dye laser was partly operated at the $Ca^+$ resonance transition. Here, 72 nights of observations are available. For $Ca^+$ it is not possible to distinguish between nights with and without sporadic layers, so all data are used.

The lidar system at the Arecibo Observatory (18° N) employed for measurements of neutral Ca at 423 nm has been described previously (Raizada et al., 2011; Raizada et al., 2012). Nd:YAG-pumped dye laser output at 701 nm is mixed with the YAG fundamental at 1064 nm to produce 423 nm light by sum frequency. The data presented in this work was collected during 2002-2003 (17 nights) and 2008 to 2010 (39 nights), giving a total of 56 nights. To eliminate the influence of strong sporadic layers, the nightly average Ca profile was fitted with a Gaussian function, which was then integrated to yield the average
column abundance.  The monthly mean abundance is used here.

For comparison with the Na layer at the respective latitudes of 54° N and 38° N we use observations from the Osiris spectrometer on the ODIN satellite (Fan et al., 2007; Dawkins et al., 2015). While these data are obtained at local times around 0600 and 1800 h in the daytime, the Na layer column abundance exhibits a very small diurnal variation (Clemesha et al., 1982) and so this data is appropriate for comparison with the nighttime Ca lidar measurements.

Nine vertical profiles of $Ca^+$ ($m/z$ =40) and $Na^+$ ($m/z$ =23) measured by rocket-borne mass spectrometry are also included. Two of these rockets were launched from Wallops Islands (37.8° N, USA) at 11:58 LT (local time) on 12[th] August 1976 and 14:03 LT on 1[st] January 1977 (payloads 18.1006 (Herrmann et al., 1978) and 18.1008 (Meister et al., 1978), respectively). Two further launches took place at Red Lake (50.9° N, Canada), at 11:52 LT  on 24[th] February 1979 and 11:55 LT on 26[th] February 1979  (payloads 18.020 and 18.021 (Kopp and Herrmann, 1984; Kopp, 1997b)). The rest were launched from Kiruna (67.8°
N, Sweden) at 01:32 LT on 30[th] July 1978 and 01:38 LT on 13[th] August 1978  (payloads S26/1 and S26/2 (Kopp et al., 1985b)),



04:50 LT on 16$^{th}$ November 1980 and 01:44 LT on 30$^{th}$ November 1980 (payloads 33.010 and 33.009 (Kopp et al., 1985a)), and at 01:32 LT on 3$^{rd}$ August 1982 (payload S37/P(Kopp et al., 1984)).

The $m/z$ =40 profile consist primarily of $Ca^+$ (Zbinden et al., 1975). Based on recent detailed modeling of Na and Mg chemistry (Plane and Whalley, 2012; Marsh et al., 2013a), significant contributions from $NaOH^+$ and $MgO^+$ can be ruled out. The 42/40

and 44/40 signal ratios in sporadic E layers have been found to be close to the $^{44}Ca/^{40}Ca$ and $^{42}Ca/^{40}Ca$ terrestrial isotopic ratios (Herrmann et al., 1978).

# 3 Results and discussion

## 3.1 Annual mean profiles of Ca species

Figure 4a compares the annual mean Ca and $Ca^+$ vertical profiles simulated by WACCM-Ca with the annual mean lidar

measurements at Kühlungsborn, and two rocket measurements of $Ca^+$ over Red Lake, Canada, which is at a similar latitude. Note that the model output was sampled to the times when lidar data was available, over the period between 1997 and 2000. The modeled and measured Ca peak height, peak density, and top and model scale-heights of the Ca layer all agree within their standard deviations. Observations show that the $Ca^+$ layer is much more variable than the Ca layer (Gerding et al., 2000; Granier et al., 1989). Although there is reasonable agreement in the general shape of the $Ca^+$ layer, and the model captures the

$Ca^+$ ion density at certain heights measured both by the lidar and the two rocket flights, overall the model appears to over-predict the $Ca^+$ density. This is a problem that we have encountered previously with WACCM modeling of both $Na^+$ (Marsh et al., 2013a) and $Fe^+$ (Feng et al., 2013), which we have attributed to the absence of electro-dynamical transport of these metallic ions in WACCM. This point is discussed further below.

Figure 4b illustrates vertical profiles of the neutral Ca species. The oxides CaO, $CaO_2$ and $CaO_3$ have relatively minor

concentrations because they are destroyed by reaction with atomic O (Table 1). The hydroxides $Ca(OH)_2$, $O_2CaOH$ and OCaOH are comparatively abundant because the H atom concentration is roughly 2 orders of magnitude lower than atomic O (Plane et al., 2015). Below the Ca layer peak (~90 km), $CaCO_3$ is the dominant gas-phase Ca species; although note that the line labelled $CaCO_3$ refers to the sum $CaCO_3 + O_2CaCO_3 + OCaCO_3$, which is mostly $O_2CaCO_3$ below 90 km, as discussed in Section 2.1. Importantly, because of the CaOH and $CaCO_3$ holding cycles, the dimerization of molecular species is much more

important than for other meteoric metals such as Fe (Feng et al., 2013) and Na (Marsh et al., 2013a). This is why the profile of the Ca sink (representing polymerized Ca molecules) is the most abundant form of the metal below 97 km. The permanent removal of Ca into these embryonic particles, and the relatively large concentrations of Ca species in the CaOH and $CaCO_3$ holding cycles, thus explains the very low abundance of neutral Ca atoms.

Figure 4c illustrates the vertical profiles of the ionized Ca species. $Ca^+$ is the dominant species above 80 km, followed by $CaO^+$

and $CaO_2^+$. This picture is quite similar to that of $Fe^+$ and its oxide ions (Feng et al., 2013). Oxidation of $Ca^+$ by $O_3$ is the



dominant reaction above 80 km (Broadley et al., 2007), because it is a bimolecular reaction in contrast to the third-order association reactions where $Ca^+$ clusters with $O_2$, $N_2$ etc. (Table 1). These association reactions are relatively slow at the low pressures in the MLT.

## 3.2 Seasonal variation of the Ca layer profile

Figure 5 displays altitude versus month plots of the Ca layer vertical profile at Kühlungsborn (54° N). Figure 5a shows the lidar measurements and Figure 5b is the standard WACCM-Ca run described in Section 2.2. Comparison of these plots, which have the same contour color scale, shows that the model simulates successfully the height and width of the layer, as well as the minimum in springtime and the broad maximum during the second half of the year. The reason for this unusual seasonal variation is that the Ca layer abundance tends to follow the seasonal variation of the Ca MIF, which peaks in autumn and has

a minimum in spring (Fig. S2 in the SI). This MIF-dependence is not really observed in the Na and Fe layers, which have a strong annual variation with a minimum in summer and maximum in winter (Feng et al., 2013; Marsh et al., 2013a). The reason for the Ca layer is strongly influenced by the MIF is because the lifetime of Ca in the layer is short: this can be estimated by dividing the Ca layer column abundance by the integrated Ca MIF, i.e. $2 \times 10^7$ cm$^{-2}$ / 420 cm$^{-2}$ s$^{-1}$, which is around 0.6 days. In comparison, the lifetime of Na atoms is much longer (around 11 days in the current simulation), so that horizontal

(particularly meridional) transport washes out the effect of the Na MIF.

The comparison between Fig. 5a and Fig. 5b shows that the peak Ca density is underestimated in July and December by around 50%. Considering that the lidar data-set is quite sparse (Gerding et al., 2000), perhaps not too much should be read into this. Nevertheless, noting that $Ca^+$ ions tend to be overestimated in the model (Fig. 4a), we carried out a sensitivity study to examine the possibility that the rate coefficients which partition calcium between Ca and $Ca^+$ were incorrectly measured. We therefore

reduced the rate coefficient $k_{24}$ for the charge transfer reaction between $NO^+$ and Ca which produces $Ca^+$. The measured rate coefficient of $4 \times 10^{-9}$ cm$^3$ molecule$^{-1}$ s$^{-1}$ (Rutherford et al., 1972) is fast compared with the Langevin capture rate (Smith, 1980) of $2.8 \times 10^{-9}$ cm$^3$ molecule$^{-1}$ s$^{-1}$. The rate coefficient $k_{25}$ for the reaction of $Ca^+$ with $O_3$, which is the most important step in converting $Ca^+$ to Ca (see Fig. 1), was also increased from its experimental value of $3.9 \times 10^{-10}$ cm$^3$ molecule$^{-1}$ s$^{-1}$ to the Langevin capture rate of $1.1 \times 10^{-9}$ cm$^3$ molecule$^{-1}$ s$^{-1}$. The result is shown in Fig. 5c. While this does have the desired effect

of increasing the Ca peak density to the observed level during July and December, at other times of the year it exceeds the measurements by more than 100%. Furthermore, the column abundance of the layer from the sensitivity simulation is substantially larger than the observations at all times of the year (not shown).

## 3.3 Seasonal variation of the Ca and Na column abundances

Figure 6a and 6b compare the modeled and measured Ca and Na column abundances as a function of season at Kühlungsborn

(54° N) and Arecibo (18° N), respectively. In both cases, there is very good agreement between the model simulations and





measurements of both metals. There are several points to note. First, there is a large annual seasonal variation (roughly a factor of 3 between the winter maximum and summer minimum) in the Na layer at 54° N, which largely disappears at tropical latitudes (18° N). Second, there is a smaller variation in the Ca layer at 54° N: the layer does not exhibit a winter maximum and summer minimum, but instead a springtime minimum and then a gradual increase during the rest of the year, which is most likely

caused by the Ca MIF increasing from spring to autumn. At 18° N, both model and measurement show almost no seasonal variation. Finally, it is important to note that the column abundances of both layers have been successfully simulated using the common factor of 5 reduction of the Ca and Na MIFs from Carrillo-Sánchez et al. (2016).

### 3.4  $Ca^+$ and $Na^+$ ratios

Figure 7a shows the geometric means of the $Ca^+$ and $Na^+$ profiles from the nine rocket profiles described in Section 2.3.

Geometric means and standard deviations are used because of the sparse data set available and the natural variability of the ion profiles. The $Ca^+$ and $Na^+$ profiles essentially overlap between 80 – 91 and 100 - 105 km.  Between 91 and 100 km, $Na^+$ is in excess by up to a factor of 6.  Figure 7a also shows the predicted ion profiles from WACCM-Ca and WACCM-Na, sampled at the locations of the rocket flights; comparisons of WACCM simulations with rocket flights at 38°, 51° and 68° N are illustrated in Fig. S4 in the SI. Note that each WACCM profile is the monthly mean around the local time of the rocket flight.

WACCM-Ca agrees with the measured $Ca^+$ profile below 82 km and above 97 km, but overestimates the $Ca^+$ density in between, though generally not by more than a factor of 2 outside the geometric standard deviation. The WACCM-Na simulation agrees less well with the rocket measurements of $Na^+$: there is reasonable agreement between 86 and 92 km, but the model then overestimates $Na^+$ by up to an order of magnitude outside the geometric standard deviation. Nevertheless, WACCM does correctly predict that the $Na^+$ excess over $Ca^+$ starts around 91 km.

More relevant for this study is the $Ca^+/Na^+$ ratio, which is illustrated in Fig. 7b. This shows that the ratio is roughly 1.0 below 90 km, and falls to 0.3 between 93 and 100 km. Note that the geometric standard deviation of the ratio is significantly smaller than the standard deviations of the individual profiles, which is expected because to some extent both ions will be subject to the same transport processes in the MLT. The modeled ratio agrees with the measured ratio between 84 and 98 km, but is too

large below 84 km and too small above 98 km. Both of these deviations appear to be largely due to the modeled $Na^+$ (Fig. 7a). The significant under-prediction of $Na^+$ below 85 km may indicate that further work on its ion-molecule chemistry is needed, and the large over-prediction above 98 km may result from differential electro-dynamical transport of the lighter $Na^+$ compared with $Ca^+$, which is not represented in WACCM (Feng et al., 2013).

Figure 7b also compares the $Ca^+/Na^+$ ratio with the CI relative abundance of these elements (red line), and the Ca/Na ablation

ratio from our recent study of the astronomical dust sources contributing to the cosmic dust input (Carrillo-Sánchez et al., 2016). This MIF ratio is depicted by the blue line. Note that the difference between the two lines, which is almost 1 order of

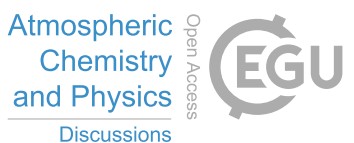

magnitude, demonstrates the differential ablation of refractory Ca compared with relatively volatile Na. A striking feature of this plot is that the rocket-measured $Ca^+/Na^+$ ratio is higher than the MIF ratio at all altitudes. This underlines the point that whereas $Ca^+$ ions are somewhat enriched relative to $Na^+$ when benchmarked against their MIFs, Ca atoms are severely depleted relative to Na by a factor of more than 100, and more than 10 against the MIF ratio.

### 3.5 Global column abundances

Figure 8a illustrates the seasonal variation of the Ca column abundance as a function of latitude. The data in this figure is listed in Table S4 in the SI. As discussed in Section 3.4 and illustrated in Fig. 6, there is good agreement between WACCM and lidar observations at $18^o$ and $54^o$ N. The original Ca layer observations were made at Observatoire d'Haute Provence ($44^o$ N), where the average column abundance of Ca between late July and December over a 5-year period (1982-87) was found to be $(2.7 \pm 1.7) \times 10^7$ $cm^{-2}$ (Granier et al., 1989). This compares well with the WACCM-Ca average of $2.4 \times 10^7$ $cm^{-2}$ during the same part of the year. Qian and Gardner (1995) made 8 nights of measurements at Urbana, Illinois ($40^oN$) between late October 1992 and January 1993. The Ca abundance on these nights ranged from $2.1 \times 10^7$ $cm^{-2}$ to $1.1 \times 10^8$ $cm^{-2}$, which encompasses the model average of $2.5 \times 10^7$ $cm^{-2}$.

In the northern hemisphere below $55^o$, the Ca abundance largely follows the Ca MIF, with a minimum in spring and a maximum in autumn (see Fig. S2). At higher latitudes, a combination of chemistry and transport causes a larger variation with a minimum in summer and maximum in winter. One important aspect is photo-ionization (reaction R38): Ca atoms have an e-folding lifetime against photo-ionization of only 5.6 hours. Thus in the summer at high latitudes which are continuously sunlit, Ca is efficiently ionized on the topside of the neutral layer. This leads to the centroid height of the layer moving down by $2-3$ km in summer at high latitudes, and the root-mean-square width of the layer decreasing by about 2 km (Fig. S3). As shown in Table 1, none of the reactions apart from R16 (which is too slow to be significant in the MLT) has a large enough activation energy to cause much temperature-dependent variation to Ca, unlike Na (Marsh et al., 2013a) and Fe (Viehl et al., 2016).

The other important factor in the high-latitude seasonal variation is the meridional transport in the MLT, which converges over the winter pole and diverges over the summer pole (Plane et al., 2015). The long-range transport of $Ca^+$ ions (Ca atoms are too short-lived, see Section 3.2) will therefore lead to an accumulation of $Ca^+$, and hence Ca, over the winter pole and *vice versa*. A similar, but more pronounced, effect has been seen in the Na and Fe layers (Gardner et al., 2005). Figure S5 in the SI contains corresponding plots to Fig. 8a for Na, illustrating this point, and also the good agreement between the measured variation of the Na column abundance using a combination of satellite and lidar data (Dawkins et al., 2015), and WACCM-Na with the Na MIF shown in Fig. 3 and revised Na chemistry (Gómez-Martín et al., 2016; Gómez-Martín et al., 2017).

Figure 8b illustrates the global variation of the $Ca^+$ column abundance. The data in this figure is listed in Table S4 in the SI. Although the interplay between transport and chemistry makes the situation in the northern hemisphere at high latitudes more





complex, dynamics seems to control the $Ca^+$ (and Ca) seasonal variation over Antarctica. The global seasonal $Ca^+$/Ca seasonal average is 11.0:1.

Figure 8c and Fig. 8d illustrate the modeled global variation of the Ca/Na and $Ca^+$/$Na^+$ ratios, respectively. The most important point to take from these figures is that the model correctly predicts the very small neutral ratio and much larger ion ratio: the

global seasonal averages are around 0.0043 and 0.14, respectively. As discussed in the Introduction, explaining the huge difference was one of the main goals of the study. Interestingly, the model predicts a small semi-annual variation in the Ca/Na ratio at all latitudes, whereas the $Ca^+$/$Na^+$ exhibits a more significant annual trend with a summer minimum and winter maximum at mid- to high-latitudes.

## 5 Conclusions

There were three objectives in the present study. The first was to incorporate the comprehensive new data-base of neutral and ion-molecule reactions of Ca pertinent to the MLT (Table 1), together with a new meteoric input function for Ca (Carrillo-Sánchez et al., 2016) that had been validated experimentally (Gómez-Martín et al., 2017), into a global chemistry-climate model. The second objective was then to explain the more than 100-fold depletion of atomic Ca relative to Na compared with their relative CI abundance; and the third was to explain why the $Ca^+$ ion abundance is depleted by only a factor of ~3 with

respect to $Na^+$ between 90 and 100 km.

In fact, these seemingly contradictory features of Ca and $Ca^+$ are linked. Ca experiences significant differential ablation, by almost 1 order of magnitude, compared to Na (Carrillo-Sánchez et al., 2016). However, Ca ionizes more efficiently than the other meteoric metals: it photo-ionizes 2.5 times faster and charge transfers 5 times faster than Na with the dominant MLT ion $NO^+$ (Plane et al., 2015). This increases the $Ca^+$/$Na^+$ ratio from the ablation ratio of 0.12 to 0.3 in the $92-100$ km height range.

In contrast with the ions, two neutral Ca species – CaOH and $CaCO_3$ – have large dipole moments and form stable oxides with $O_2$. Although $O_2$CaOH and $O_2CaCO_3$ recycle back to CaOH and $CaCO_3$ via reactions with O above the atomic O shelf around 82 km, they then rapidly recombine again with $O_2$, thus forming stable holding cycles. We have shown this experimentally in the case of CaOH (Gómez-Martín and Plane, 2017), and for $CaCO_3$ in the present study using electronic structure theory coupled with statistical rate theory. Most of the neutral Ca is therefore held in the CaOH and $CaCO_3$ holding cycles, enabling

rapid loss through polymerization into meteoric smoke particles. This explains the unexpectedly low abundance of neutral Ca atoms in the MLT, which was first observed more than 30 years ago (Granier et al., 1985).



**Author contribution**

JMCP carried out the theoretical calculations and chemical model development. WF designed the WACCM model experiments and performed the simulations. JCGM contributed to the Ca kinetic scheme and analyzed the rocket data. MG and SR processed the lidar data. JMCP prepared the manuscript with contributions from all co-authors.

**Acknowledgments and Data**

This work was supported by the European Research Council (project 291332 - CODITA). The rocket flight data was kindly provided by E. Kopp (University of Bern). The WACCM model and data input are available through a subversion repository at svn-ccsm-release.cgd.ucar.edu. The MERRA-2 data sets used for the specified dynamics in WACCM were provided by the Climate Data Gateway at National Center for Atmospheric Research (www.earthsystemgrid.org). The WACCM, lidar and
rocket data data sets generated for this work have been archived at the Leeds University PetaByte Environmental Tape Archive and Library (PETAL; http://www.see.leeds.ac.uk/business-and-consultation/ facilities/petabyte-environmental-tapearchive-and-library-petal/). The electronic structure theory calculations, additional WACCM-Ca results and a table of the Ca global column abundance climatology are included in the supplemental information.

The authors declare that they have no conflict of interest

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





## Table 1. Gas-phase Neutral and Ion-molecule Reactions of Ca species

| Number | Reaction | Rate Coefficient[a] |
|---|---|---|
| | *Neutral Chemistry* | |
| R1 | $Ca + O_3 \rightarrow CaO + O_2$ | $8.2 \times 10^{-10} \exp(-192/T)$ [b] |
| R2 | $Ca + O_2(a^1\Delta_g) \rightarrow CaO + O$ | $2.7 \times 10^{-12}$ [c] |
| R3 | $CaO + O \rightarrow Ca + O_2$ | $1.1e \times 10^{-9} \exp(-421/T)$ [d] |
| R4 | $CaO + O_3 \rightarrow CaO_2 + O_2$ | $5.7 \times 10^{-10} \exp(-267/T)$ [e] |
| R5 | $Ca + O_2 (+ M) \rightarrow CaO_2$ | $1.2 \times 10^{-30} (T/200)^{3.65}$ [f] |
| R6 | $CaO_2 + O \rightarrow CaO + O_2$ | $4.4 \times 10^{-11} \exp(-202/T)$ [d] |
| R7 | $CaO_2 + O_3 \rightarrow CaO_3 + O_2$ | $1 \times 10^{-10} (T/200)^{0.5}$ [g] |
| R8 | $CaO_2 + H \rightarrow CaOH + O$ | $1.2 \times 10^{-11}$ [d] |
| R9 | $CaO + H_2O (+ M) \rightarrow Ca(OH)_2$ | $7.3 \times 10^{-25} (T/200)^{-2.12}$ [e] |
| R10 | $CaO + O_2 (+ M) \rightarrow CaO_3$ | $6.4 \times 10^{-28} (T/200)^{-0.358}$ [e] |
| R11 | $CaO + CO_2 (+ M) \rightarrow CaCO_3$ | $2.9 \times 10^{-27} (T/200)^{-1.07}$ [e] |
| R12 | $CaO_3 + H_2O \rightarrow Ca(OH)_2 + O_2$ | $5 \times 10^{-12}$ [g] |
| R13 | $CaO_3 + CO_2 \rightarrow CaCO_3 + O_2$ | $5 \times 10^{-12}$ [g] |
| R14 | $CaO_3 + O \rightarrow CaO_2 + O_2$ | $2 \times 10^{-11}$ [g] |
| R15 | $CaO_3 + H \rightarrow CaOH + O_2$ | $1.7 \times 10^{-11}$ [d] |
| R16 | $CaCO_3 + O \rightarrow CaO_2 + CO_2$ | $4.0 \times 10^{-12} \exp(-4689/T)$ [d,h] |
| R17 | $Ca(OH)_2 + H \rightarrow CaOH + H_2O$ | $1 \times 10^{-11}$ [d] |
| R18 | $CaOH + H \rightarrow Ca + H_2O$ | $1.0 \times 10^{-10}$ [i] |
| R19 | $CaOH + O_2 (+M) \rightarrow O_2CaOH$ | $k_0 = 8.9 \times 10^{-26} (300/T)^{4.99}$ [i] $k_\infty = 1.5 \times 10^{-10} (T/300)^{0.167}$ $F_c = 0.136$ |
| R20 | $O_2CaOH + O \rightarrow OCaOH + O_2$ | $2 \times 10^{-10}$ [i] |
| R21 | $OCaOH + O \rightarrow CaOH + O_2$ | $1.5 \times 10^{-10}$ [i] |
| R22 | Polymerization of CaOH, $Ca(OH)_2$, $OCaO_2H$, $OCaOH$, $CaCO_3$ | $9 \times 10^{-8}$ [j] |
| | *Ion-molecule Chemistry* | |
| R23 | $Ca + O_2^+ \rightarrow Ca^+ + O_2$ | $1.8 \times 10^{-9}$ [k] |
| R24 | $Ca + NO^+ \rightarrow Ca^+ + NO$ | $4.0 \times 10^{-9}$ [k,l] |



| | | | |
|---|---|---|---|
| R25 | $Ca^+ + O_3 \rightarrow CaO^+ + O_2$ | $3.9 \times 10^{-10}$ [m,n] | |
| R26 | $CaO^+ + O \rightarrow Ca^+ + O_2$ | $4.2 \times 10^{-11}$ [o] | |
| R27 | $Ca^+ + O_2 (+ M) \rightarrow CaO_2^+$ | $4.2 \times 10^{-29} (T/200)^{-2.37}$ [m] | |
| R28 | $CaO_2^+ + O \rightarrow CaO^+ + O_2$ | $1.0 \times 10^{-10}$ [o] | |
| R29 | $Ca^+ + N_2 + M \rightarrow CaN_2^+ + M$ | $2.3 \times 10^{-30} (T\,200)^{-2.49}$ [m] | 5 |
| R30 | $Ca^+ + CO_2 (+ M) \rightarrow Ca^+.CO_2$ | $4.3 \times 10^{-29} (T/200)^{-3.09}$ [m] | |
| R31 | $Ca^+ + H_2O (+ M) \rightarrow Ca^+.H_2O$ | $1.2 \times 10^{-28} (T/200)^{-2.12}$ [m] | |
| R32 | $CaN_2^+ + O_2 \rightarrow CaO_2^+ + N_2$ | $3 \times 10^{-10}$ [o] | |
| R33 | $Ca^+.CO_2 + O_2 \rightarrow CaO_2^+ + CO_2$ | $1.2 \times 10^{-10}$ [o] | 10 |
| R34 | $Ca^+.CO_2 + H_2O \rightarrow Ca^+.H_2O + CO_2$ | $1.3 \times 10^{-9}$ [o] | |
| R35 | $Ca^+.H_2O + O_2 \rightarrow CaO_2^+ + H_2O$ | $4.0 \times 10^{-10}$ [o] | |
| R36 | $CaX^+ + e^- \rightarrow Ca + X$ | $3 \times 10^{-7} (T/295)^{-1/2}$ [p] | 15 |
| | $(X = O, \ O_2, N_2, CO_2, H_2O)$ | | |
| R37 | $Ca^+ + e^- \rightarrow Ca + h\nu$ | $3.8 \times 10^{-12} (T/200)^{-0.9}$ [q] | |
| | *Photochemical Reactions* | | 20 |
| R38 | $Ca + h\nu \rightarrow Ca^+ + e^-$ | $5 \times 10^{-5}$ [r] | |

[a] Units: unimolecular, $s^{-1}$; bimolecular, $cm^3$ molecule$^{-1}$ $s^{-1}$; termolecular, $cm^6$ molecule$^{-2}$ $s^{-1}$. Rate coefficients are from: [b] Helmer et al. (1993); [c] Plane et al. (2012); [d] Broadley and Plane (2010); [e] Plane and Rollason (2001); [f] Campbell and Plane (2001); [g] Estimate. [h] Calculated using transition state theory (see the SI); [i] Gómez-Martín and Plane (2017); [j] Fitted sink reaction (see text); [k] Rutherford et al. (1972); [l] in the sensitivity run, reduced to the Langevin capture rate of $2.8 \times 10^{-9}$ $cm^3$ $s^{-1}$ (see text); [m] Broadley et al. (2007); [n] in the sensitivity run, increased to the Langevin capture rate of $1.1 \times 10^{-9}$ $cm^3$ $s^{-1}$ (see text); [o] Broadley et al. (2008); [p] Bones et al. (2016a); [q] Shull and van Steenberg (1982); [r] Calculated from the Ca photo-ionization cross section (McIlrath and Sandeman 1972; Ahmad et al., 1994).





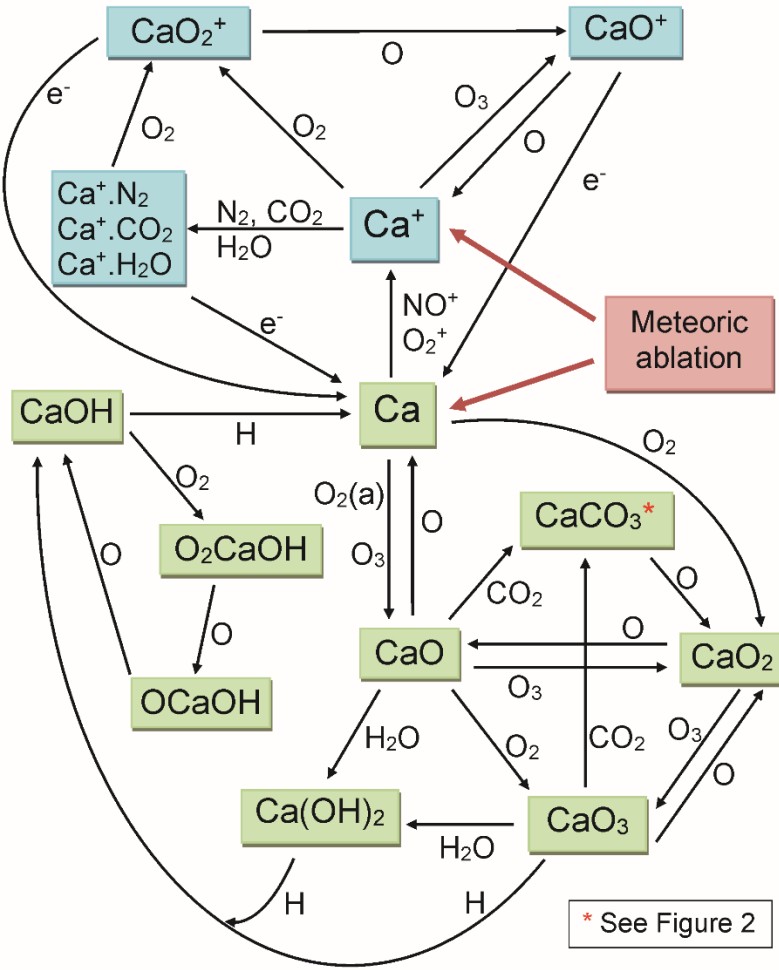

**Figure 1: Schematic diagram of the Ca chemistry in WACCM-Ca. Ionized and neutral Ca-containing species are shown in blue and green boxes, respectively.**





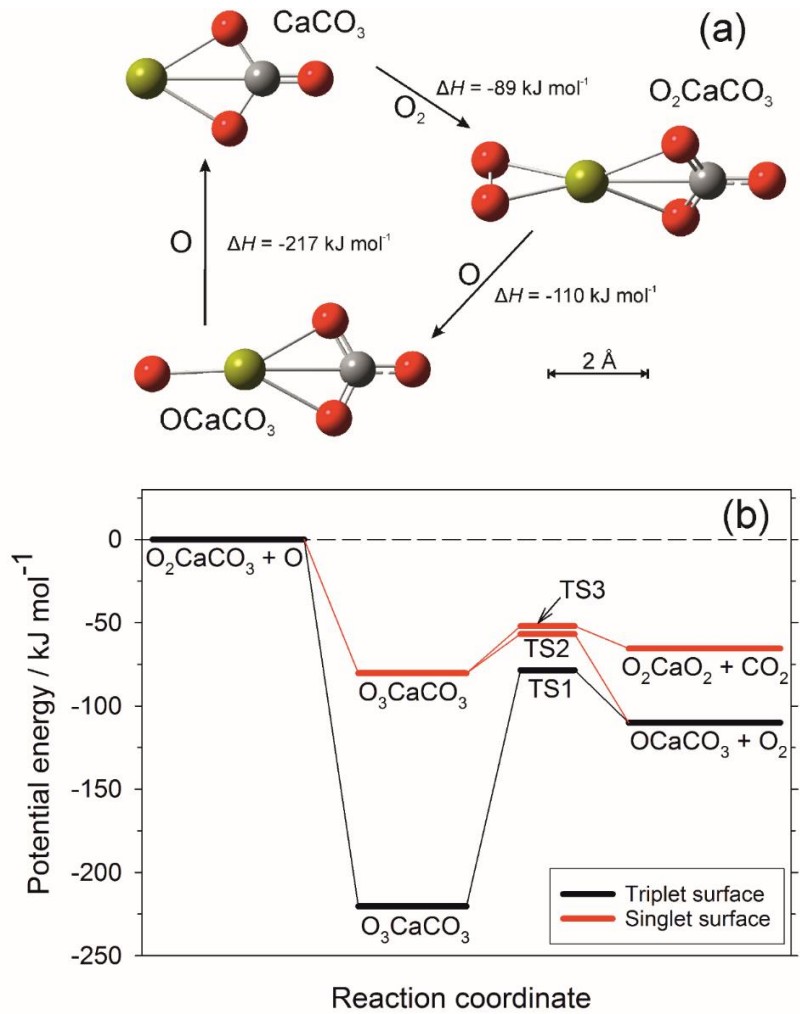

Figure 2: (a) The CaCO₃ holding cycle, analogous to that of CaOH (Gómez-Martín and Plane, 2017). Colour scheme: Ca (yellow); C (grey); O (red). The reaction enthalpies (at 0 K) for each step were calculated at the B3LYP/6-311+g(2d,p) level of theory (Frisch et al., 2009). (b) Potential energy surface for the reaction between O₂CaCO₃ and O, calculated at the B3LYP/6-311+g(2d,p) level of theory. Zero-point energies are included in the energies of the stationary points. The quintet surface for this reaction is non-reactive.



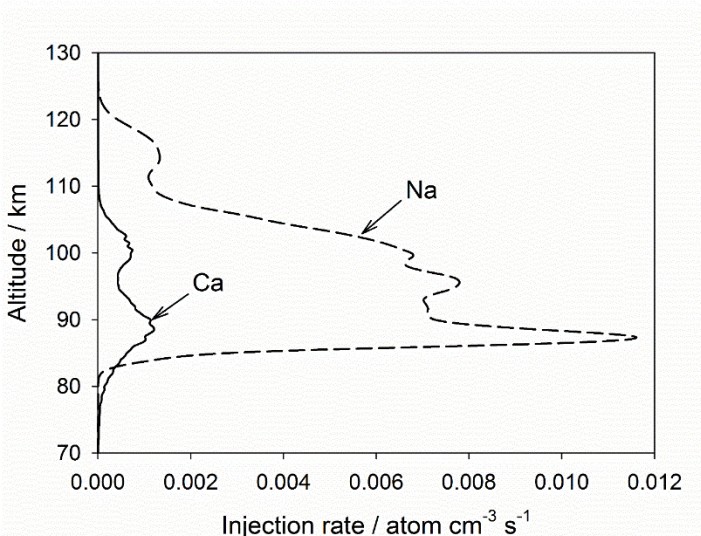

**Figure 3: Global annual mean injection rate profiles of Ca and Na resulting from meteoric ablation, used in WACCM. Note that these injection rates have been reduced by a factor of 5 from those determined by *Carrillo-Sanchez et al.* (2016).**



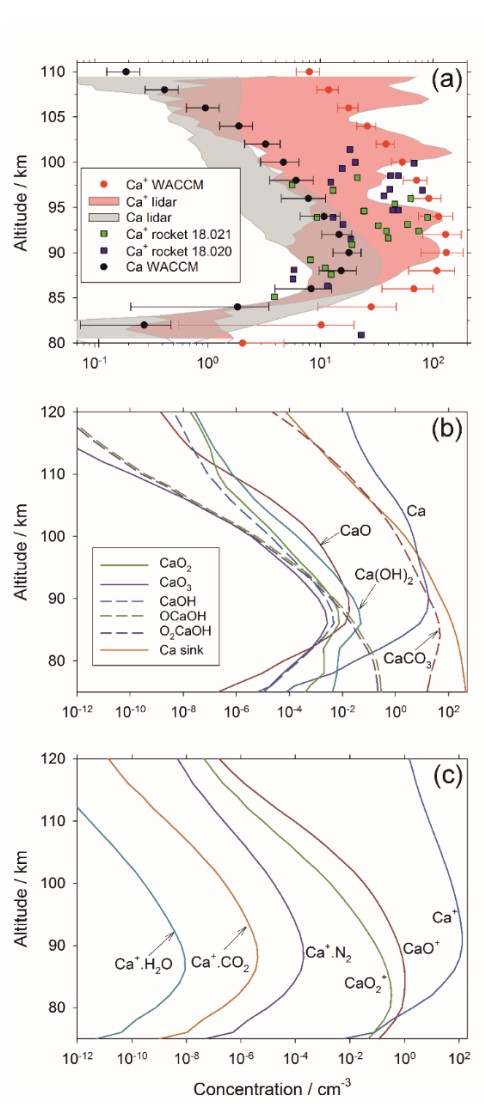

**Figure 4: Annual mean concentration profiles of Ca species over Kühlungsborn (54° N): (a) comparison of WACCM-Ca model results (the horizontal error bars indicate 1 standard deviation from the mean) with the lidar measurements of Ca and Ca⁺ (the shaded areas encompass 1 standard deviation from the mean). Note that the model output was sampled to the lidar data over the same period between 1997 and 2000. Also shown are mass spectrometric measurements of Ca⁺ made by two rockets which were launched from Red Lake (50.9° N, Canada), on 24/Feb/1979 11:52 LT and 26/Feb/1979 11:55 LT (payloads 18.020 and 18.021 (Kopp and Herrmann, 1984; Kopp, 1997b)); (b) modelled vertical profiles of neutral Ca species in WACCM-Ca, where "CaCO₃" = CaCO₃ + O₂CaCO₃ + OCaCO₃; (c) modelled vertical profiles of ionic Ca species in WACCM-Ca.**





**Figure 5: Annual variation of the Ca layer over Kühlungsborn (54° N): (a) lidar measurements; (b) WACCM-Ca standard model run; (c). WACCM-Ca sensitivity run where the rate coefficients for R24 and R25 are set to their respective Langevin capture rates.**



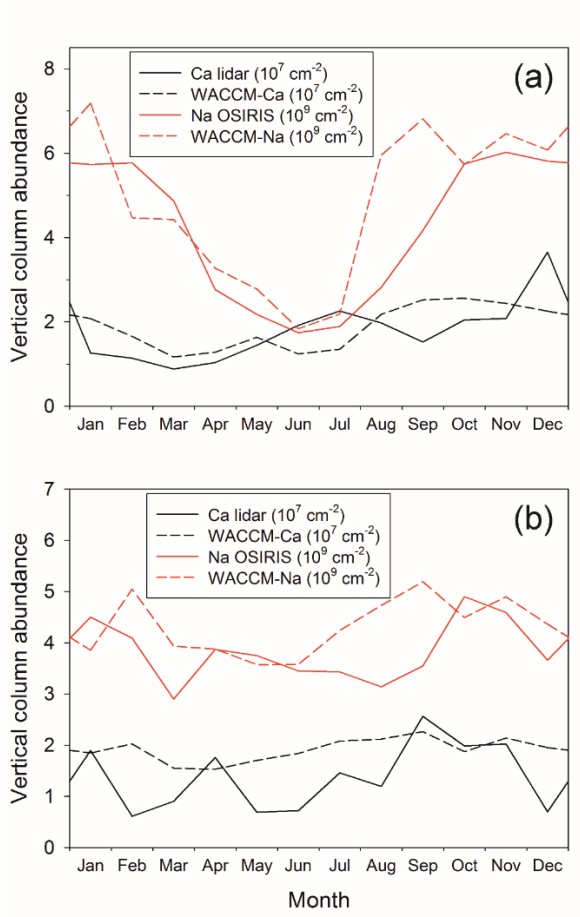

**Figure 6: Comparison of observed and modelled Ca and Na vertical column abundances: (a) at Kühlungsborn (54° N); (b) at Arecibo (18° N). The Na column observations were made with the Osiris spectrometer on the ODIN satellite (Fan et al., 2007).**



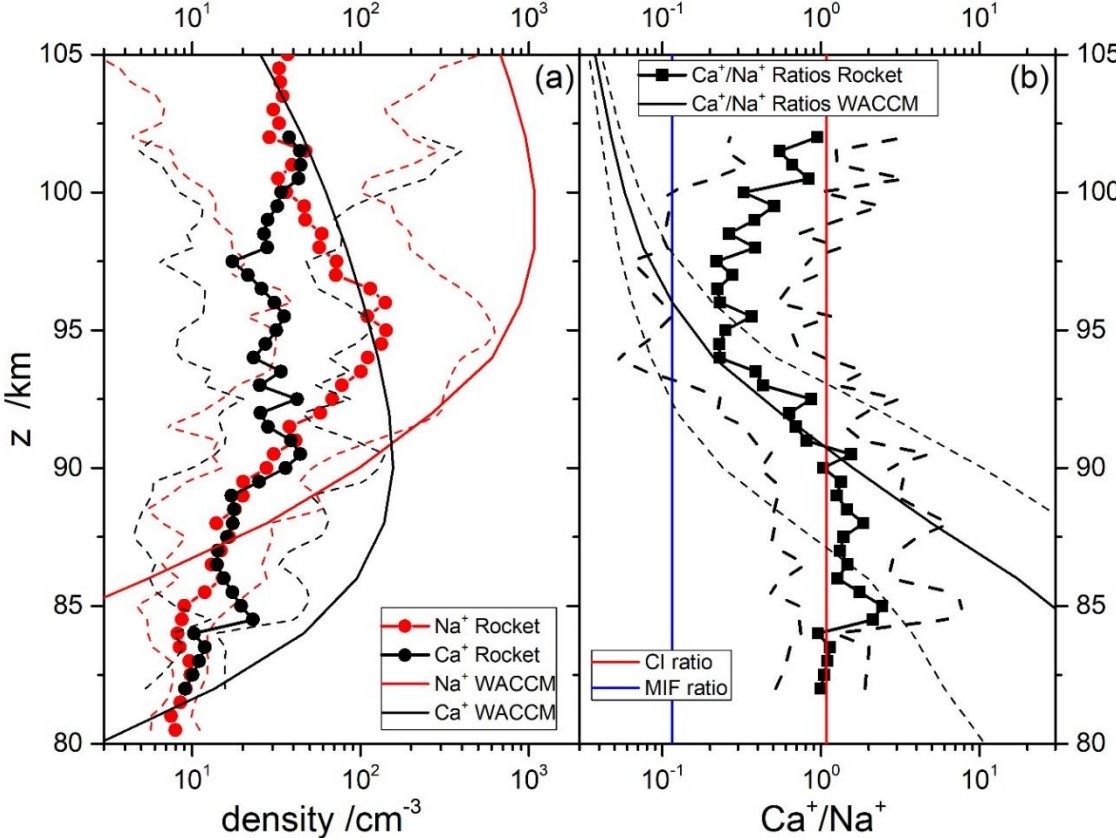

**Figure 7: Comparison of observed and modelled Ca⁺ and Na⁺ density profiles: (a) geometric mean profiles of rocket-borne mass spectrometric measurements of Ca⁺ and Na⁺ (circles) with geometric standard deviations (dashed lines), and WACCM model (thick solid lines); (b) geometric mean profiles of the observed (squares) and modelled (thick solid lines) Ca⁺/Na⁺ ratios, with geometric standard deviations (dashed lines). The Ca/Na abundance ratio in carbonaceous Ivuna (CI) chondrites (Hutchison, 2004) and the ratio of meteor input functions (Carrillo-Sánchez et al., 2016) are indicated with red and blue lines, respectively.**




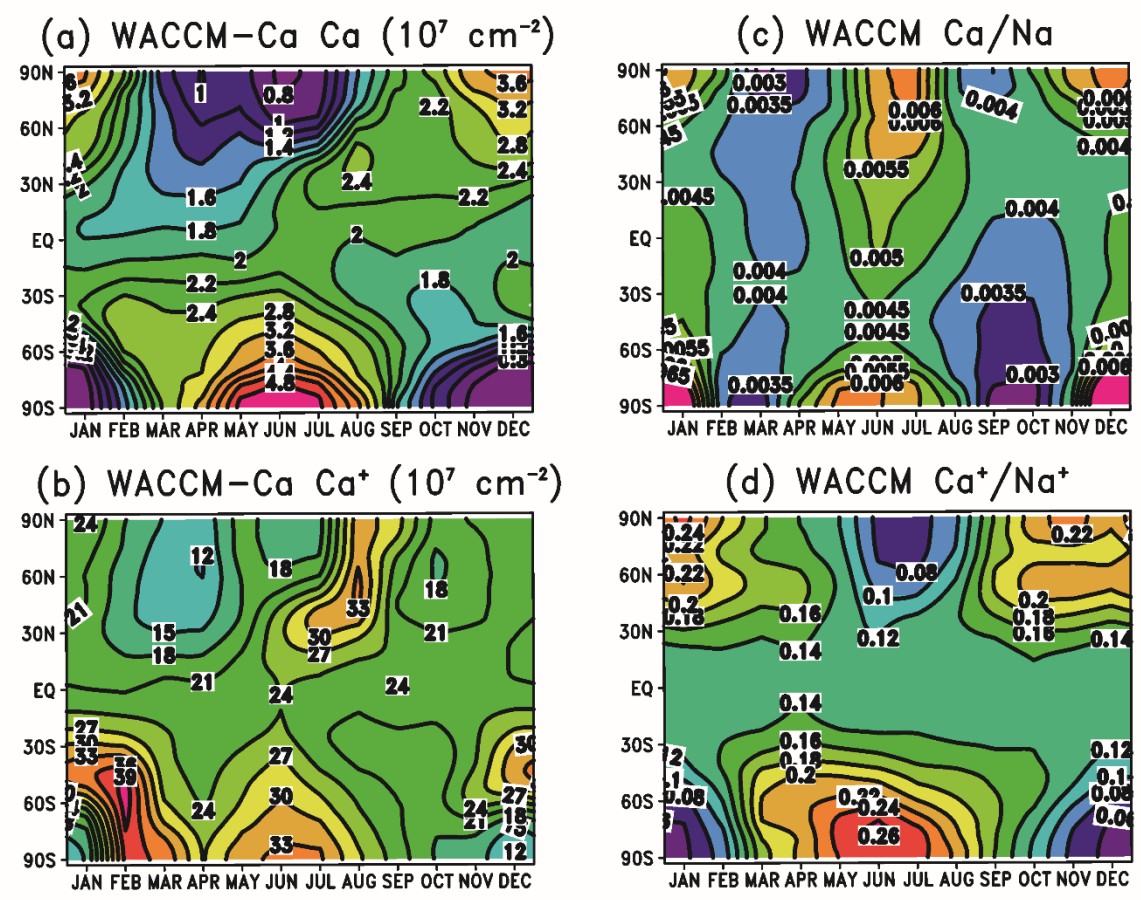

**Figure 8: Diurnally averaged column abundances predicted by WACCM, plotted as a function of latitude and month: (a) Ca; (b) Ca⁺; (c) ratio of Ca/Na; and (d) ratio of Ca⁺/Na⁺.**

