# Peer review of "A new model of meteoric calcium in the mesosphere and lower thermosphere"

_Atmospheric Chemistry and Physics, 2018_

## Referee Comment (RC1) · Anonymous Referee #1 · 18 Jun 2018

This is the latest in a series of papers, led by the University of Leeds group, which describe the results of global modeling of the meteoric metals. This paper addresses the detailed chemistry and modeling of mesospheric Ca and Ca+ using the Whole Atmosphere Community Climate Model (WACCM) developed at the U.S. National Center for Atmospheric Research. This is perhaps the most important test of these WACCM metal models as it explains quite convincingly the extremely low abundance of Ca relative to Na, as well as the seemingly contradictory large Ca+/Na+ ratio. Na and Ca have similar elemental abundances in chondritic meteorites but the measured Ca abundance in the mesosphere and lower thermosphere (MLT) is depleted by more than a factor of 100 relative to Na while the Ca+ abundance is depleted by less than a factor of 10. The authors show quantitatively that these contradictory observations are a consequence

of substantial differences in the ablation rates, ionization rates and chemical loss rates of Ca and Na. The detailed chemistry is described and the model is then compared with lidar measurements of Ca and Ca+ at Kühlungsborn and Arecibo and rocket measurements of Ca+ and Na+ at several sites. The agreement between the model and observations is impressive. This is an important contribution to our understanding of the metal chemistry and of the impact of dynamical transport on these species in the MLT.

The paper is succinct but informative and exceptionally well written. The figures are all essential and well crafted and the text is adequately referenced. I recommend the paper be published as is.
* * *

---

## Referee Comment (RC2) · Anonymous Referee #2 · 29 Sep 2018

This manuscript describes WACCM simulations of Ca, Ca+ density variations, based on substantially improved Ca reaction kinetics and experimentally-based Meteoric Input Function, and comparisons with lidar and rocket measurements. The goal is to find an explanation for the extremely large depletion of Ca compared to Na and the unusually large Ca+/Ca ratio. Based on the new understanding of the chemistry, it is proposed that this is due to very stable CaOH and CaCO3 which act as a reservoir of Ca.

The manuscript is well-written, with a very clear explanation of the proposed questions, and they are well supported by the WACCM simulations and observations. I recommend it to be published as is, and only have a few minor questions/suggestions listed below.

[Figure]

P1, L19-20, and P11, L16-17: These sentences do not state which ablation is larger, Ca or Na. It's better to state that explicitly, although it is mentioned in P3, L4. The sentence can be simply changed to "... 1 order of magnitude larger than Na"

P4, L16-17: Please explain why 100 times increase is used instead of some other factor.

P5, L18: 200 km is a closer to the WACCM resolution at 1.9 degrees.

P11, L2: It's simpler to just use '11.0' instead of '11.0:1'

---

## Author Comment (AC1) · 29 Sep 2018

**Title**: A new model of meteoric calcium in the mesosphere and lower thermosphere

**Author(s)**: John M. C. Plane et al.

**MS No.**: acp-2018-493

**Response to the reviewers' comments**

We thank both reviewers for their very positive reviews of the paper. The points raised by Reviewer #2 are listed below in italics, with our response in normal type.

*P1, L19-20, and P11, L16-17: These sentences do not state which ablation is larger, Ca or Na. It's better to state that explicitly, although it is mentioned in P3, L4. The sentence can be simply changed to "… 1 order of magnitude larger than Na"*

The sentence in the abstract (P1) has been changed to: "A new meteoric input function for Ca and Na, derived using a chemical ablation model that has been tested experimentally with a Meteoric Ablation Simulator, shows that Ca ablates almost 1 order of magnitude less efficiently than Na."

The sentence on P11 has been changed to: "Ca ablates almost 1 order of magnitude less efficiently than Na".

*P4, L16-17: Please explain why 100 times increase is used instead of some other factor.*

This is now explained on P4, L18: "The reason for increasing the rate coefficient by a factor of ~100 is that the Ca reservoir species can also polymerize with other (i.e., non-Ca containing) meteoric molecules (e.g., $NaHCO_3$, $FeOH$, and $Mg(OH)_2$), and the dimerization rate coefficient needs to be increased to account for this since Ca ablates in a large excess of these other metals: the elemental ablation ratio of Ca atoms to the sum of Na, Fe, Mg, Si, Al and K atoms is 0.01 (Carrillo-Sánchez et al., 2016)."

*P5, L18: 200 km is a closer to the WACCM resolution at 1.9 degrees.*

Agreed – now changed.

*P11, L2: It's simpler to just use '11.0' instead of '11.0:1'*

Agreed – now changed.